# A Video Is Worth 4096 Tokens:
# Verbalize Videos To Understand Them In Zero Shot

**Aanisha Bhattacharyya**⋆  **Yaman K Singla**⋆

**Balaji Krishnamurthy**  **Rajiv Ratn Shah**  **Changyou Chen**

Adobe Media and Data Science Research (MDSR),  IIIT-Delhi,  State University of New York at Buffalo

## Abstract

Multimedia content, such as advertisements and story videos, exhibit a rich blend of creativity and multiple modalities. They incorporate elements like text, visuals, audio, and storytelling techniques, employing devices like emotions, symbolism, and slogans to convey meaning. There is a dearth of large annotated training datasets in the multimedia domain hindering the development of supervised learning models with satisfactory performance for real-world applications. On the other hand, the rise of large language models (LLMs) has witnessed remarkable zero-shot performance in various natural language processing (NLP) tasks, such as emotion classification, question-answering, and topic classification. To leverage such advanced techniques to bridge this performance gap in multimedia understanding, we propose verbalizing long videos to generate their descriptions in natural language, followed by performing video-understanding tasks on the generated story as opposed to the original video. Through extensive experiments on fifteen video-understanding tasks, we demonstrate that our method, despite being zero-shot, achieves significantly better results than supervised baselines for video understanding. Furthermore, to alleviate a lack of story understanding benchmarks, we publicly release the first dataset on a crucial task in computational social science on persuasion strategy identification.

## 1  Introduction

*"We are, as a species, addicted to stories. Even when the body goes to sleep, the mind stays up all night, telling itself stories."* - Jonathan Gottschall

Most videos we encounter in the day-to-day, like movies, documentaries, advertisements, and user-generated content like Tiktok and Youtube shorts,

depict some form of a story. Despite this, most work in the multimedia understanding domain has been about simple videos containing a single action or photo streams (Li et al., 2020). Beyond understanding objects, actions, and scenes lies interpreting causal structure, making sense of visual, textual, and audio input to tie disparate moments together as they give rise to a cohesive narrative of events through time. This requires moving from reasoning about single activity and static moments to sequences of images and audio that depict events as they occur and change. Progressing from single-action videos to story videos allows us to begin to reason about complex cognitive tasks like emotions depicted and persuasion strategies used.

Recently, large video pre-trained models (LVMs) like VideoMAE (Tong et al., 2022), InternVideo (Wang et al., 2022), and VideoCLIP (Xu et al., 2021) have proved to be powerful in enhancing reasoning skills on video data. For *e.g.*, InternVideo showed a performance increase in action classification and question answering tasks. Nevertheless, these models have a few shortcomings that impair their performance on video understanding tasks: 1) LVMs are mostly trained on short videos (<10s) consisting of majorly motion-centric actions, such as those present in Kinetics (Kay et al., 2017) and Something Something v2 (Goyal et al., 2017); and 2) they often require a significant amount of task-specific finetuning data to perform on video-understanding tasks like summarization, question-answering, and emotion classification. On the other hand, stories are often longer than 10 seconds and are typically much more complex than motion-centric videos. For example, we test our models on five datasets with average video lengths of 12.4 minutes (video story) and 3.5 minutes (other tasks). Our videos contain dialogues, text overlaid on frames, fast-moving scenes, graphics like symbols and cartoon characters, and rhetoric elements like emotions and taglines, other than motions and

---
⋆Equal Contribution. Contact ykumar@adobe.com for questions and suggestions.

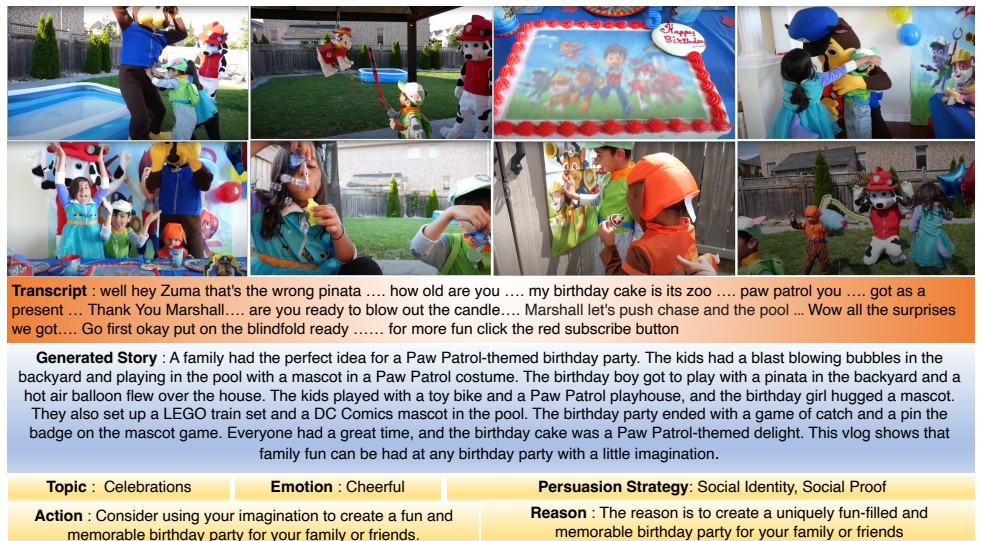

Transcript : well hey Zuma that's the wrong pinata …. how old are you …. my birthday cake is its zoo …. paw patrol you …. got as a present … Thank You Marshall…. are you ready to blow out the candle…. Marshall let's push chase and the pool ... Wow all the surprises we got…. Go first okay put on the blindfold ready …… for more fun click the red subscribe button

Generated Story : A family had the perfect idea for a Paw Patrol-themed birthday party. The kids had a blast blowing bubbles in the backyard and playing in the pool with a mascot in a Paw Patrol costume. The birthday boy got to play with a pinata in the backyard and a hot air balloon flew over the house. The kids played with a toy bike and a Paw Patrol playhouse, and the birthday girl hugged a mascot. They also set up a LEGO train set and a DC Comics mascot in the pool. The birthday party ended with a game of catch and a pin the badge on the mascot game. Everyone had a great time, and the birthday cake was a Paw Patrol-themed delight. This vlog shows that family fun can be had at any birthday party with a little imagination.

Topic : Celebrations | Emotion : Cheerful | Persuasion Strategy: Social Identity, Social Proof

Action : Consider using your imagination to create a fun and memorable birthday party for your family or friends. | Reason : The reason is to create a uniquely fun-filled and memorable birthday party for your family or friends

Figure 1: An example of a story generated by the proposed pipeline along with the predicted outputs of the video-understanding tasks on the generated story. The generated story captures information across scenes, characters, event sequences, dialogues, emotions, and the environment. This helps the downstream models to get adequate information about the video to reason about it correctly. The original video can be watched at https://youtu.be/_amwPjAcoC8.

actions. Further, in the video domain, there are not large enough datasets for finetuning LLMs on the downstream tasks.

Recently, zero-shot performance in the natural language processing (NLP) domain has increased substantially owing to the recent growth of generative large language models (LLMs). Instead of fine-tuning LLMs, in-context learning has recently gained noticeable attention to exploring the reasoning ability of LLM, where several input-output exemplars are provided for prompting (Wei et al., 2022; Kojima et al., 2022; Min et al., 2022). For example, the chain of thought prompting (Wei et al., 2022) has been discovered to empower LLMs to perform complex reasoning by generating intermediate reasoning steps.

Owing to in-context learning, tasks like emotion recognition, named entity recognition (Wang et al., 2023), and even higher-order composite tasks like table understanding (Ye et al., 2023) have shown performance improvements. However, the capability of LLMs on video reasoning tasks is still unexplored. There are several technical challenges preventing leveraging LLMs for video-based reasoning tasks. First, a video in the raw form can be quite long, spanning minutes and thus containing many frames. Directly encoding all frames via pre-trained models could be computationally intractable and interfere with huge amounts of irrelevant information. Second, for effective video understanding, we need information about multi-

ple sources and modalities such as dialogue, text, characters, and scenes. All of them provide both intersecting and mutually exclusive information helpful to understand and reason about the video. As a first work, we propose to address these challenges by exploiting the power of LLMs in the video domain. The LLMs are used to decompose long videos into stories and then to reason about videos using the generated stories instead of the raw information. In this way, we can retain the relevant evidence and exclude the remaining irrelevant evidence from interfering with the decision.

The main contributions of our paper are:

1) We propose converting long videos from the multimodal domain to "small" coherent *textual* stories by verbalizing keyframes, audio, and text-overlaid scenes with the help of a powerful LLM and instructions (Figs. 1, 5). We experimentally test the story generation capability of our method and note that our method outperforms state-of-the-art story generation methods.

2) Next, we test the utility of generated stories by conducting extensive experiments on five benchmark datasets covering fifteen video understanding tasks. Experimental results demonstrate that our methods achieve better results than both fine-tuned and zero-shot video understanding baseline models without using any human-annotated samples. Through this, for the first time in literature, we show that the essence of a highly-multimodal video can be represented in text while being in-

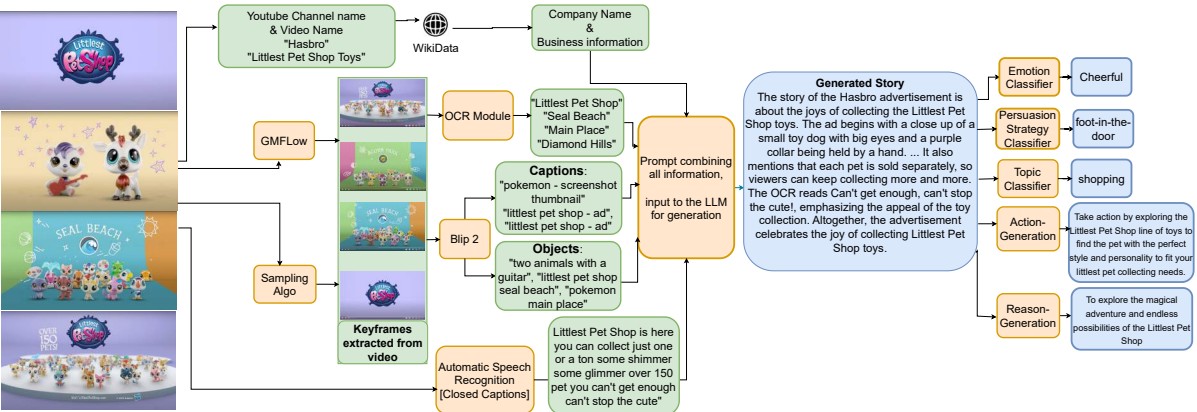

Figure 2: The overview of our framework to generate a story from a video and perform downstream video-understanding tasks. First, we sample keyframes from the video which are verbalized using BLIP-2. We also extract OCR from all the frames. Next, using the channel name and ID, we query Wikidata to get company and product information. Next, we obtain automatically generated captions from Youtube videos using the Youtube API. All of these are concatenated as a single prompt and given as input to an LLM and ask it to generate the story of the advertisement. Using the generated story, we then perform the downstream tasks of emotion and topic classification and persuasion strategy identification. This video can be watched at `https://youtu.be/ZBLkTALi1CI`.

formed through the different modalities like audio, raw pixels of frames, text overlaid on scenes, emotions, and product and business information. This text representation can then be used to perform story-understanding tasks instead of the original video. We show that finetuning on video stories rather than videos leads to better performance (Table 2). Through an ablation study, we also show that none of the individual modalities is able to perform as well as the combined knowledge when crystallized in a story using LLMs (Table 8).

3) Further, given the lack of datasets for complex story video-understanding tasks, we release the first dataset for studying persuasion strategies in advertisement videos (Fig. 3)[1]. This enables initial progress on the challenging task of automatically understanding the messaging strategies conveyed through video advertisements.

## 2 Related Work

**Visual Storytelling:** Bridging language and multimedia is a longstanding goal in multimedia understanding (Mogadala et al., 2021). Earlier works mainly target image and video captioning tasks, where a single sentence factual description is generated for an image or a segment (Xu et al., 2015). Recent works (Krause et al., 2017; Liang et al., 2017; Yu et al., 2016; Krishna et al., 2017; Li et al., 2020) aim to provide more comprehensive and fine-

grained descriptions by generating multi-sentence paragraphs. However, most work is concentrated on cooking videos and action videos. Li et al. (2020) is arguably the first to propose a dataset for long videos with complex event dynamics, aiming to generate a coherent and succinct story from the abundant and complex visual data. We use their dataset to test our method. Further, recently, two datasets specifically targeting long-form video understanding were released: Long Video Understanding (LVU) (Wu and Krahenbuhl, 2021) and Holistic Video Understanding (HVU) (Diba et al., 2020). In consonance with the idea of stories expressed in long-form videos, these datasets go beyond actions and objects and release rich semantic labels like concepts, attributes, and events in HVU, and relationship, scene, way of speaking, *etc.* in the LVU dataset. We use these datasets to test long-video understanding.

**Advertisement Story Understanding:** Other than user-generated content, the other most important source of story content is brand-generated content. Through these videos, brands try to communicate with their customers. There has been some work on understanding advertisements (Hussain et al., 2017; Ye and Kovashka, 2018; Zhang et al., 2018; Ye et al., 2019; Savchenko et al., 2020; Pilli et al., 2020; Gaikwad et al., 2022; Kumar et al., 2023). Hussain et al. (2017) released a dataset containing image and video advertisements and their emotion, topic, action-reason, symbolism, and other labels. We show our performance on all the

---

[1]Visit `https://github.com/midas-research/video-persuasion` to access the videos and their annotations.

video tasks they released in their work. Subsequent papers improve performance benchmarks on their image ads dataset.

**Understanding Persuasion In Ad Stories:** Further, we contribute one crucial task on the advertisement story understanding task: persuasion strategy identification. The primary purpose of all brand communication is to change people's beliefs and actions (*i.e.* to persuade). There has been limited work in computer vision on persuasion. Among the limited prior works, Bai et al. (2021) tried to answer the question of which image is more persuasive; and Joo et al. (2014) introduced syntactical and intent features such as facial displays, gestures, emotion, and personality, which result in persuasive images. On the other hand, decoding persuasion in textual content has been extensively studied in natural language processing from both extractive and generative contexts (Habernal and Gurevych, 2016; Chen and Yang, 2021; Luu et al., 2019). All of the marketing messages employ one of a set of strategies to persuade their target customers. Kumar et al. (2023) came up with an exhaustive list of strategies used by brands to persuade consumers and also released an image-based ads dataset annotated with those strategies. We build on that list and annotate video ads for persuasion strategies.

**Large Language Models on Reasoning:** Large language models (LLMs) have been shown to confer a range of reasoning abilities, such as arithmetic (Lewkowycz et al., 2022), commonsense (Liu et al., 2022), and symbolic reasoning (Zhou et al., 2022), as the model parameters are scaled up (Brown et al., 2020). Notably, chain-of-thought (CoT) (Wei et al., 2022) leverages a series of intermediate reasoning steps, achieving better reasoning performance on complex tasks. Building on this, Kojima et al. (2022) improved reasoning performance by simply adding "Let's think step by step" before each answer. Fu et al. (2022) proposed generating more reasoning steps for the chain to achieve better performance. Zhang et al. (2022) proposed selecting examples of in-context automatically by clustering without the need for manual writing. Despite the remarkable performance of LLMs in textual reasoning, their reasoning capabilities on video-understanding tasks are still limited.

## 3 Method

Large Language Models (LLMs) have been demonstrated to perform well for downstream classification tasks in the text domain. This powerful ability has been widely verified on natural language tasks, including text classification, semantic parsing, mathematical reasoning, *etc*. Inspired by these advances of LLMs, we aim to explore whether they could tackle reasoning tasks on multimodal data (*i.e.* videos). Therefore, we propose a storytelling framework, which leverages the power of LLMs to verbalize videos in terms of a text-based story and then performs downstream video understanding tasks on the generated story instead of the original video. Our pipeline can be used to verbalize videos and understand videos to perform complex downstream tasks such as emotion, topic, and persuasion strategy detection.

### 3.1 Video Verbalization

To obtain a verbal representation of a video, we employ a series of modules that extract unimodal information from the multimodal video. This information is then used to prompt a generative language model (such as GPT-3.5 (Brown et al., 2020) and Flan-t5 (Chung et al., 2022)) to generate a coherent narrative from the video. The overall pipeline is depicted in Fig. 2. In the following, we delve into each component of the framework in details.

**1. Video Metadata:** Understanding the context of a story is crucial, and we achieve this by gathering information about the communicator (brand). We leverage the publicly available video title and channel name from the web. Additionally, we utilize Wikidata (Vrandečić and Krötzsch, 2014), a collaborative knowledge base that provides comprehensive data for Wikipedia, to obtain further details such as the company name, product line, and description. This information helps us comprehend the story elements and establish connections with the brand's business context. For non-advertisement videos, we skip this step and retrieve only the video title.

**2. Text Representation of Video Frames:** We extract two types of textual information from video frames. Firstly, we capture the literal text present on the frames. Secondly, we analyze the scene depicted in each frame to gain a deeper understanding. In the upcoming sections, we will elaborate on both of these aspects.

*a. Visual and Scenic Elements in Frames:* For videos with a duration shorter than 120 seconds, we employ an optical flow-based heuristic using the GMFlow model (Xu et al., 2022) to extract

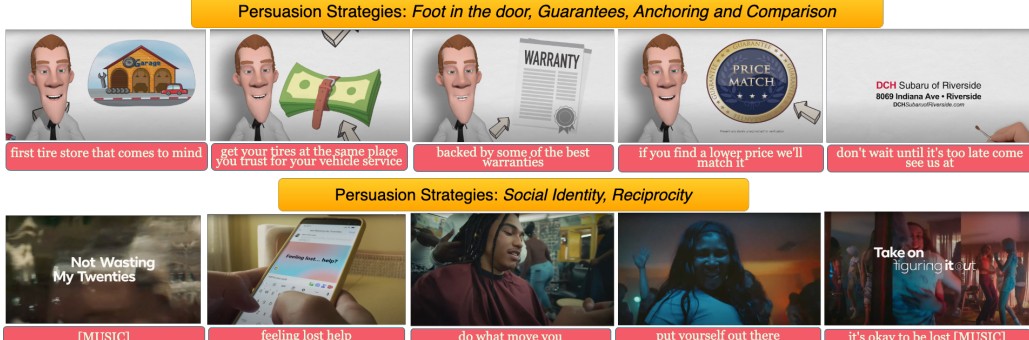

Figure 3: Examples of videos with their annotated persuasion strategies. Relevant keyframes and ASR captions are shown in the figure, along with the annotated strategies. These two videos can be watched at https://bit.ly/3Ie3JG0, https://bit.ly/3OgtLwj .

keyframes. In shorter advertisement videos, scene changes often indicate transitions in the story, resulting in keyframes with higher optical flow values. The GMFlow model effectively captures these story transitions. We select frames with an optical flow greater than 50 and prioritize frames with maximum pixel velocity. However, for longer videos, this approach yields a large number of frames that are difficult to accommodate within a limited context. To address this, we sample frames at a uniform rate based on the native frames-per-second (fps) of the video (see Table 11 for a comparison between uniform sampling and Pyscenedetect). Additionally, we discard frames that are completely dark or white, as they may have high optical flow but lack informative content.

Using either of these methods, we obtain a set of frames that represent the events in the video. These frames are then processed by a pretrained BLIP-2 model (Li et al., 2023a). The BLIP model facilitates scene understanding and verbalizes the scene by capturing its most salient aspects. We utilize two different prompts to extract salient information from the frames. The first prompt, "*Caption this image*," is used to generate a caption that describes what is happening in the image, providing an understanding of the scene. The second prompt, "*Can you tell the objects that are present in the image?*," helps identify and gather information about the objects depicted in each frame.

*b. Textual elements in frames:* We also extract the textual information present in the frames, as text often reinforces the message present in a scene and can also inform viewers on what to expect next (Wang et al., 2021). For the OCR module, we sample every 10th frame extracted at the native frames-per-second of the video, and these frames are sent to PP-OCR (Vrandečić and Krötzsch, 2014). We filter the OCR text and use only the unique words for further processing.

**3. Text Representation of Audio:** The next modality we utilize from the video is the audio content extracted from it. We employ an Automatic Speech Recognition (ASR) module to extract transcripts from the audio. Since the datasets we worked with involved YouTube videos, we utilized the YouTube API to extract the closed caption transcripts associated with those videos.

**4. Prompting:** We employ the aforementioned modules to extract textual representations of various modalities present in a video. This ensures that we capture the audio, visual, text, and outside knowledge aspects of the video. Once the raw text is collected and processed, we utilize it to prompt a generative language model in order to generate a coherent story that represents the video. To optimize the prompting process and enable the generation of more detailed stories, we remove similar frame captions and optical character recognition (OCR) outputs, thereby reducing the overall prompt size.

The prompt template is given in Appendix A.1.2. Through experimentation, we discovered that using concise, succinct instructions and appending the text input signals (such as frame captions, OCR, and automatic speech recognition) at the end significantly enhances the quality of video story generation. For shorter videos (up to 120 seconds), we utilize all available information to prompt the LLM for story generation. However, for longer videos, we limit the prompts to closed captions and sampled frame captions. The entire prompting pipeline is zero-shot and relies on pre-trained LLMs. In our story generation experiments, we employ GPT-3.5 (Brown et al., 2020), Flan-t5 (Chung et al., 2022),

and Vicuna (Chiang et al., 2023). A temperature of 0.75 is used for LLM generation. The average length of the generated stories is 231.67 words. Subsequently, these generated stories are utilized for performing video understanding tasks.

## 3.2 Downstream Video Understanding Tasks

For each downstream video understanding task, we explain the task in detail to the LLMs and provide the options to choose from. We adopt an in-context learning approach by using task descriptions as prompts . This method enables the LLMs to acquire the skills necessary to solve the understanding task effectively. The provided context encompasses comprehensive information about the classification task, including details about different classes, along with their corresponding definitions and concepts. By exposing the language model to this context, it becomes adept at selecting the correct option when presented with a generated text input.

In the story generation pipeline illustrated in Fig. 2, we leverage the stories derived from videos and employ separate prompting systems to classify tags for downstream tasks. Each task operates independently, ensuring that the context of one task does not interfere with another. Here too, we utilize three generative language models, namely GPT-3.5, Flan-t5, and Vicuna. In these experiments, we employ a lower temperature setting of 0.3. Our performance is evaluated against various fine-tuned models from the literature. Moreover, we utilize our generated stories and ground truth labels to perform fine-tuning on a Roberta model for various tasks. Both Vicuna and GPT-3.5 generated stories are used to train the Roberta model. This allows us to compare the performance of models fine-tuned on video data with those trained solely on video-verbalized-text data.

## 4 Evaluation and Results

### 4.1 Datasets

To test the effectiveness of our framework, we conduct experiments involving fifteen distinct tasks across five datasets. Firstly, we employ a video story dataset to evaluate the story generation task. Secondly, we utilize a video advertisements dataset to assess topic and emotion classification, as well as action and reason generation. Then, the persuasion strategy dataset to evaluate the task of understanding persuasion strategies within stories, and finally, HVU and LVU for concept, user engagement, and

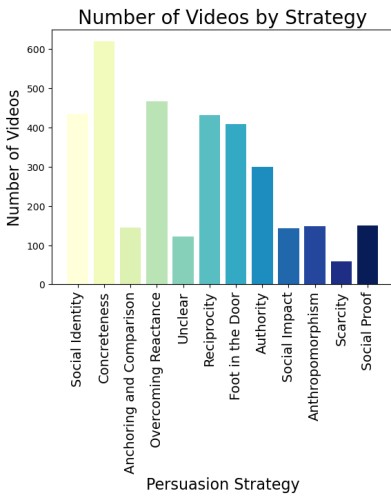

Figure 4: Distribution of persuasion strategies in our dataset

attribute prediction. These diverse datasets allow us to evaluate the performance and capabilities of our framework thoroughly.

**1. The Video story dataset** (Li et al., 2020) contains 105 videos, from four types of common and complex events (*i.e.* birthday, camping, Christmas, and wedding) and corresponding stories written by annotators. It has longer videos (average length 12.4 mins) and longer descriptions (162.6 words on average). Moreover, the sentences in the dataset are more sparsely distributed across the video (55.77 sec per sentence). *Metrics*: Following Li et al. (2020), we use several NLP metrics, *viz.*, BLEU-N, ROUGE-L, METEOR and CIDEr to measure the similarity between the story generated by the model and ground truth.

**2. The Image and Video Advertisements** (Hussain et al., 2017) contains 3,477 video advertisements and the corresponding annotations for emotion and topic tags and action-reason statements for each video. There are a total of 38 topics and 30 unique emotion tags per video. Further, we have 5 action-reason statements for each video for the action-reason generation task. For our experiment, we use 1785 videos, due to other videos being unavailable/privated from Youtube.

*Metrics*: Following Hussain et al. (2017), for the topic and emotion classification task, we evaluate our pipeline using top-1 accuracy as the evaluation metric. Further, since Hussain et al. (2017) did not use any fixed set of vocabulary for annotations, rather they relied on annotator-provided labels, the labels are often very close (like cheerful, excited, and happy). Therefore, based on near-

| | Method | Model Type | METEOR | CIDEr | ROUGE-L | BLEU-1 | BLEU-2 | BLEU-3 | BLEU-4 |
|---|---|---|---|---|---|---|---|---|---|
| Random | Random | Retrieval | 13.1 | 30.2 | 21.4 | 43.1 | 23.1 | 10.0 | 4.8 |
| **Finetuned** | Narrator (Li et al., 2020) | Retrieval | 19.6 | 98.4 | 29.5 | 69.1 | 43.0 | 25.3 | 15.0 |
| | EMB (Li et al., 2020) | Retrieval | 19.1 | 88.8 | 28.9 | 64.5 | 39.3 | 22.7 | 13.4 |
| | BRNN (Li et al., 2020) | Retrieval | 18.1 | 81.0 | 28.3 | 61.4 | 36.6 | 20.3 | 11.3 |
| | ResBRNN (Li et al., 2020) | Retrieval | 19.6 | 94.3 | 29.7 | 66.0 | 41.7 | 24.3 | 14.7 |
| | Pseudo-GT+ ResBRNN-kNN (Li et al., 2020) | Retrieval | 20.1 | 103.6 | 29.9 | 69.1 | 43.5 | 26.1 | 15.6 |
| | GVMF (Lu and Wu, 2022) | Retrieval | 20.7 | 107.7 | 30.8 | 70.5 | 44.3 | 26.9 | 15.9 |
| **Zero-shot** | VideoChat (Li et al., 2023b) | Generative | 15.49 | 42.9 | 17.88 | 50.00 | 43.30 | 34.76 | 27.21 |
| **Zero-shot** | GPT-3.5 | Generative | 24.8 | 102.4 | 24.3 | 63.8 | 56.4 | 47.2 | 38.6 |
| Our Framework | Vicuna | Generative | 17.4 | 73.9 | 20.9 | 70.49 | 60.0 | 48.25 | 38.20 |
| | Flant-t5-xxl | Generative | 4.8 | 34.6 | 10.58 | 7.9 | 6.8 | 5.4 | 4.3 |
| | Uniformly Sampled BLIP-2 Captions | Generative | 21.7 | 108.9 | 24.04 | 55.19 | 48.5 | 40.7 | 33.76 |

Table 1: Comparison on story generation task on the video-story dataset. We see that our framework despite being zero-shot outperforms all the fine-tuned generative prior art on all metrics. Further, it also outperforms fine-tuned retrieval models, which choose from a fixed set of frame descriptions on most metrics. Best models are denoted in green and runner-ups in blue .

| Training | Model | Topic | Emotion | | Persuasion | Action | Reason |
|---|---|---|---|---|---|---|---|
| | | | All labels | Clubbed | | | |
| Random | Random | 2.63 | 3.37 | 14.3 | 8.37 | 3.34 | 3.34 |
| Finetuned | VideoMAE (Tong et al., 2022) | 24.72 | 29.72 | 85.55 | 11.17 | - | - |
| | Hussain et al. (2017) | 35.1 | 32.8 | - | - | - | 48.45 |
| | Intern-Video (Wang et al., 2022) | 57.47 | 36.08 | 86.59 | 5.47 | 6.8 | 7.1 |
| Zero-shot | VideoChat (Li et al., 2023b) | 9.07 | 3.09 | 5.1 | 10.28 | - | - |
| **Our Framework** | GPT-3.5 Generated Story + GPT-3.5 Classifier | 51.6 | 11.68 | 79.69 | 35.02 | 66.27 | 59.59 |
| Zero-shot | GPT-3.5 Generated Story + Flan-t5-xxl Classifier | 60.5 | 10.8 | 79.10 | 33.41 | 79.22 | 81.72 |
| | GPT-3.5 Generated Story + Vicuna Classifier | 22.92 | 10.8 | 67.35 | 29.6 | 21.39 | 20.89 |
| | Vicuna Generated Story + GPT-3.5 Classifier | 46.7 | 5.9 | 80.33 | 27.54 | 61.88 | 55.44 |
| | Vicuna Generated Story + Flan-t5-xxl Classifier | 57.38 | 9.8 | 76.60 | 30.11 | 77.38 | 80.66 |
| | Vicuna Generated Story + Vicuna Classifier | 11.75 | 10.5 | 68.13 | 26.59 | 20.72 | 21.00 |
| Finetuned | Generated Story + Roberta Classifier | 71.3 | 33.02 | 84.20 | 64.67 | 42.96[1] | 39.09[1] |

Table 2: Comparison of all the models across topic, emotion, and persuasion strategy detection tasks. We see that our framework, despite being zero-shot, outperforms finetuned video-based models on the topic classification, persuasion strategy detection and action and reason classification tasks and comes close on the emotion classification task. Further, the Roberta classifier trained on generated stories outperforms both finetuned and zero-shot models on most tasks. Best models are denoted in green and runner-ups in blue .

ness in Plutchik (1980) wheel of emotions, we club nearby emotions and use these seven main categories: joy, trust, fear, anger, disgust, anticipation, and unclear. For the action-reason task, following Hussain et al. (2017), we evaluate our accuracy on the action and reason retrieval tasks where 29 random options along with 1 ground truth are provided to the model to find which one is the ground truth. Further, we also generate action and reason statements and evaluate the generation's faithfulness with the ground truth using metrics like ROUGE, BLEU, CIDEr, and METEOR.

**3. Persuasion strategy dataset**: Further, we contribute an important task on advertisement story understanding task, namely persuasion strategy identification. Fig. 3 shows a few examples from the curated dataset. For this task, we collected 2203 video advertisements from popular brands available on the web publicly and use the persuasion strategy labels defined by Kumar et al. (2023). We use the following 12 strategies as our target persua-

sion strategy set: *Social Identity, Concreteness, Anchoring and Comparison, Overcoming Reactance, Reciprocity, Foot-in-the-Door, Authority, Social Impact, Anthropomorphism, Scarcity, Social Proof,* and *Unclear*. In order to make the class labels easier to understand for non-expert human annotators, we make a list of 15 yes/no type-questions containing questions like "*Was there any expert (person or company) (not celebrity) encouraging to use the product/brand? Was the company showcasing any awards (e.g., industrial or government)? Did the video show any customer reviews or testimonials?*" (complete list in Appendix:Table 6).

Each human annotator watches 15 videos such that each video gets viewed by at least two annotators and answers these questions for each video. Based on all the responses for a video, we assign labels to that video. We remove videos with an inter-annotator score of less than 60%. After removing those, we get a dataset with 1002 videos, with an average length of 33 secs and a distribution

| Task | Model | METEOR | CIDEr | ROUGE-L | BLEU-1 | BLEU-2 | BLEU-3 | BLEU-4 |
|---|---|---|---|---|---|---|---|---|
| Action | GPT-3.5 | 20.46 | 41.7 | 9.5 | 18.7 | 14.8 | 11.8 | 9.4 |
| Action | Flan-t5-xxl | 15.75 | 61.5 | 13.6 | 50.0 | 34.8 | 26.9 | 21.8 |
| Action | Vicuna | 21.20 | 42.6 | 7.6 | 16.8 | 13.08 | 10.08 | 7.7 |
| Reason | GPT-3.5 | 13.34 | 16.7 | 7.8 | 27.1 | 20.8 | 14.7 | 10.4 |
| Reason | Flan-t5-xxl | 8.35 | 24.9 | 5.9 | 39.4 | 24.7 | 16.7 | 12.0 |
| Reason | Vicuna | 15.82 | 27.9 | 7.75 | 24.6 | 19.3 | 14.1 | 10.3 |
| Reason given action | GPT-3.5 | 13.77 | 29.4 | 8.7 | 33.5 | 24.9 | 17.9 | 13.2 |
| Reason given action | Flan-t5-xxl | 4.29 | 19.0 | 7.6 | 23.2 | 15.0 | 10.2 | 7.5 |
| Reason given action | Vicuna | 13.62 | 24.4 | 7.61 | 22.6 | 17.7 | 12.8 | 9.2 |

Table 3: Comparison of the different zero-shot models on the action and reason generation tasks. Note that there are no fine-tuned generative models in the literature for this task and the number of annotated videos is too small to train a generative model. Best models are denoted in green .

as shown in Fig. 4. This dataset is then used for the persuasion strategy identification task. *Metrics*: We evaluate the performance using top-1 accuracy metric. Videos have a varied number of strategies, therefore, we consider a response to be correct if the predicted strategy is present among the list of ground-truth strategies.

**4. Long-Form Video Understanding (LVU):** Wu and Krahenbuhl (2021) released a benchmark comprising of 9 diverse tasks for long video understanding and consisting of over 1000 hours of video. The various tasks consist of content understanding ('relationship', 'speaking style', 'scene/place'), user engagement prediction ('YouTube like ratio', 'YouTube popularity'), and movie metadata prediction ('director', 'genre', 'writer', 'movie release year'). Wu and Krahenbuhl (2021) use top-1 classification accuracy for content understanding and metadata prediction tasks and MSE for user engagement prediction tasks.

**5. Holistic Video Understanding (HVU):** HVU (Diba et al., 2020) is the largest long video understanding dataset consisting of 476k, 31k, and 65k samples in train, val, and test sets, respectively. A comprehensive spectrum includes the identification of various semantic elements within videos, consisting of classifications of scenes, objects, actions, events, attributes, and concepts. To measure performance on HVU tasks, similar to the original paper, we use the mean average precision (mAP) metric on the validation set.

### 4.2 Results

**Video Storytelling:** The performance comparison between our pipeline and existing methods is presented in Table 1. We evaluate multiple generative and retrieval-based approaches and find that our pipeline achieves state-of-the-art results. It is important to note that as our method is entirely gen-

erative, the ROUGE-L score is lower compared to retrieval-based methods due to less overlap with ground truth reference video stories. However, overall metrics indicate that our generated stories exhibit a higher level of similarity to the reference stories and effectively capture the meaning of the source video.

**Video Understanding:** The performance comparison between our pipeline and other existing methods across six tasks (topic, emotion, and persuasion strategy classification, as well as action and reason retrieval and generation) is presented in Tables 2 and 3. Notably, our zero-shot model outperforms finetuned video-based baselines in all tasks except emotion classification. Further, our text-based finetuned model outperforms all other baselines on most of the tasks.

Unlike the story generation task, there are limited baselines available for video understanding tasks. Moreover, insufficient samples hinder training models from scratch. To address this, we utilize state-of-the-art video understanding models, Video-MAE and InternVideo. InternVideo shows strong performance on many downstream tasks. Analyzing the results, we observe that while GPT-3.5 and Vicuna perform similarly for story generation (Table 1), GPT-3.5 and Flan-t5 excel in downstream tasks (Table 2). Interestingly, although GPT-3.5 and Vicuna-generated stories yield comparable results, GPT-3.5 exhibits higher performance across most tasks. Vicuna-generated stories closely follow GPT-3.5 in terms of downstream task performance.

Next, we compare the best models (as in Table 2) on the LVU and HVU benchmarks with respect to the state-of-the-art models reported in the literature. Tables 4 and 5 report the results for the comparisons. As can be noted, the zero-shot models outperform most other baselines. For LVU,

| Training | Model | relationship | way_speaking | scene | like_ratio | view_count | director | genre | writer | year |
|---|---|---|---|---|---|---|---|---|---|---|
| Trained | R101-slowfast+NL (Wu and Krahenbuhl, 2021) | 52.4 | 35.8 | 54.7 | 0.386 | 3.77 | 44.9 | 53.0 | 36.3 | 52.5 |
| Trained | VideoBert (Sun et al., 2019) | 52.8 | 37.9 | 54.9 | 0.320 | 4.46 | 47.3 | 51.9 | 38.5 | 36.1 |
| Trained | Xiao et al. (2022) | 50.95 | 34.07 | 44.19 | 0.353 | 4.886 | 40.19 | 48.11 | 31.43 | 29.65 |
| Trained | Qian et al. (2021) | 50.95 | 32.86 | 32.56 | 0.444 | 4.600 | 37.76 | 48.17 | 27.26 | 25.31 |
| Trained | Object Transformers (Wu and Krahenbuhl, 2021) | 53.1 | 39.4 | 56.9 | 0.230 | 3.55 | 51.2 | 54.6 | 34.5 | 39.1 |
| Zero-shot (Ours) | GPT-3.5 generated story + Flan-t5-xxl | 64.1 | 39.07 | 60.2 | 0.061 | 12.84 | 69.9 | 58.1 | 52.4 | 75.6 |
| Zero-shot (Ours) | GPT-3.5 generated story + GPT-3.5 classifier | 68.42 | 32.95 | 54.54 | 0.031 | 12.69 | 75.26 | 50.84 | 32.16 | 75.96 |
| Trained (Ours) | GPT-3.5 generated story + Roberta | 62.16 | 38.41 | 68.65 | 0.054 | 11.84 | 45.34 | 39.27 | 35.93 | 7.826 |

Table 4: Comparison of various models on the LVU benchmark. We see that our framework, despite being zero-shot, outperforms fine-tuned video-based models on 8/9 tasks. Best models are denoted in green and runner-ups in blue .

| Training | Model | Scene | Object | Action | Event | Attribute | Concept | Overall |
|---|---|---|---|---|---|---|---|---|
| Trained | 3D-Resnet | 50.6 | 28.6 | 48.2 | 35.9 | 29 | 22.5 | 35.8 |
| Trained | 3D-STCNet | 51.9 | 30.1 | 50.3 | 35.8 | 29.9 | 22.7 | 36.7 |
| Trained | HATNet | 55.8 | 34.2 | 51.8 | 38.5 | 33.6 | 26.1 | 40 |
| Trained | 3D-Resnet (Multitask) | 51.7 | 29.6 | 48.9 | 36.6 | 31.1 | 24.1 | 37 |
| Trained | HATNet (Multitask) | 57.2 | 35.1 | 53.5 | 39.8 | 34.9 | 27.3 | 41.3 |
| Zero-Shot (Ours) | GPT-3.5 generated story + Flan-t5-xxl classifier | 59.66 | 98.89 | 98.96 | 38.42 | 67.76 | 86.99 | 75.12 |
| Zero-Shot (Ours) | GPT-3.5 generated story + GPT-3.5 classifier | 60.2 | 99.16 | 98.72 | 40.79 | 67.17 | 88.6 | 75.77 |

Table 5: Comparison of various models on the HVU benchmark (Diba et al., 2020). The models scores are as reported in Diba et al. (2020). We see that our framework, despite being zero-shot, outperforms fine-tuned video-based models on all the tasks. Best models are denoted in green and runner-ups in blue .

the zero-shot models work better than the trained Roberta-based classifier model. For HVU, we convert the classification task to a retrieval task, where in a zero-shot way, we input the verbalization of a video along with 30 randomly chosen tags containing an equal number of tags for each category (scene, object, action, event, attribute, and concept). The model is then prompted to pick the top 5 tags that seem most relevant to the video. These tags are mapped back to the main category tags, which are treated as the predicted labels.

Furthermore, as a comparative and ablation study of our approach, we evaluate the performance using only the BLIP-2 captions and audio transcriptions (Table 8). Our findings highlight that generated stories leveraging both audio and visual signals outperform those using vision or audio inputs alone. This emphasizes the significance of verbalizing a video in enhancing video understanding.

## 5    Conclusion

In this study, we delve into the challenge of understanding long videos. These videos, including advertisements and documentaries, encompass a wide range of creative and multimodal elements, such as text, music, dialogues, visual scenery, emotions, and symbolism. However, the availability of annotated benchmark datasets for training models from scratch is limited, posing a significant obstacle. To overcome these challenges, we propose leveraging the advancements in large language models (LLMs) within the field of natural language processing (NLP). LLMs have shown remarkable zero-shot accuracy in various text-understanding tasks. Thus, our approach involves verbalizing videos to generate stories and performing video understanding on these generated stories in a zero-shot manner. We utilize signals from different modalities, such as automatic speech recognition, visual scene descriptions, company names, and scene optical character recognition, to prompt the LLMs and generate coherent stories. Subsequently, we employ these stories to understand video content by providing task explanations and options. Our proposed method demonstrates its effectiveness across fifteen video-understanding tasks getting state-of-the-art results across many of them. The entire pipeline operates in a zero-shot manner, eliminating the reliance on dataset size and annotation quality.

**Acknowledgement**: This work is partially supported by NSF AI Institute-2229873, NSF RI-2223292, an Amazon research award, a Google fellowship, and an Adobe gift fund. Any opinions, findings, conclusions, or recommendations expressed in this material are those of the author(s) and do not necessarily reflect the views of the National Science Foundation, the Institute of Education Sciences, or the U.S. Department of Education.

# 6 Limitations

In this work, we showed that verbalizing videos results in better performance on downstream tasks when compared to finetuned video based computer vision models. We verbalized videos using video transcripts, scene captions, optical character recognition, and other techniques. The overall performance of the pipeline is intricately linked to the output generated to its constituent components. However, it is important to acknowledge the potential for hallucinations arising from each component, which can introduce false information and imaginative outputs. The final output's hallucinations and human values alignment are dependent on the underlying LLM, which need more research. We have tried to give a few qualitative samples of the same in the Appendix.

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

# A Appendix

## A.1 Experimentation Details

Video and Brands selection - We covered 155 Fortune 500 brands, covering 113 industries, the ads spanned the years 2008-2022. The videos have an avg duration of 33 secs. Nearly 45% of the videos have audio in them. The collected advertisement videos have a variety of characteristics, including different scene velocities, human presence and animations, visual and audio branding, a variety of emotions, visual and scene complexity, and audio types. The total number of unique annotators is 484. Annotators were university students who were given guidelines to annotate videos and were duly compensated for their work based on local rates. We use the following 12 strategies as our target persuasion strategy set: *Social Identity, Concreteness, Anchoring and Comparison, Overcoming Reactance, Reciprocity, Foot-in-the-Door, Authority, Social Impact, Anthropomorphism, Scarcity, Social Proof,* and *Unclear*. In order to make the class labels easier to understand for non-expert human annotators, we make a list of 15 yes/no type-questions containing questions that are mentioned below.

### A.1.1 Sampling Rate

The frames for longer videos (>2 mins), every 10th frame at the native fps of the video is sampled and similar captions are deduplicated to reduce the context size for Prompting. Refer Table 11 for a comparison between uniform sampling and Pyscenedetect.

### A.1.2 Prompt format

For verbalization, a template prompt format has been used, including all the data components as objects, captions, asr, ocr, meta-data.

*"Please write a coherent story based on the following video advertisement. Use only the information provided and make sure the story feels like a continuous narrative and at the end include one sentence about what product the advertisement was about. Do not include any details not mentioned in the prompt.Use the elements given below to create a coherent narrative,but don't use them as it is.The advertisement for the company {company_name} The video is titled {title}, with captions that include {caption}, voice-over : {transcripts}, and object recognition descriptions : {ocr}. The following objects are present in the advertisement and should be used to help create the story: {objects} Please*

*exclude any empty or stop words from the final text."*

For downstream tasks, a template prompt format with an instruction about the specific task,the previous generated verbalization and vocabulary for the downstream task is prompted to the LLM. Here is the example for the topic detection task, for other tasks context and vocab were changed accordingly.

*"Given {topics} identify the most relevant topic from the dictionary keys from topic_vocab related to the story of the video advertisement given below.Consider the definitions given with topics in the topic_vocab dictionary, to identify which topic is most relevant, don't add any extra topics that are not given in dictionary keys and answer with just the most relevant topic. Story : {verbalization}"*

## A.2 A few examples of the stories generated using our method

1. https://www.youtube.com/watch?v=lPdD8NvVfw0: Kathy Ames had always wanted to pursue a doctoral degree but was unsure about the time commitment. When she discovered Grand Canyon University, she knew she had found the perfect fit. Grand Canyon University offered a flexible schedule that would allow her to balance her personal and family life with her studies. She - along with other students - gathered in the classroom, excitedly listening to their coach, Scott Saunders, explain the program. Afterward, Kathy made her way to the library and settled into a chair with her laptop.

She studied diligently, surrounded by her peers and classmates. In the evenings, she met with her peers around the table to discuss the topics of the day. Everyone was always eager to help and support each other. After a long day, Kathy made her way back to her living room where she relaxed on the couch with a glass of water and a lamp providing a soothing light.

Kathy was grateful for the opportunity to pursue her dream at Grand Canyon University. She was able to learn from experienced faculty and gain real-world experience that would prepare her for success after graduation.

The advertisement for Grand Canyon University was about offering a private, Christian education at an affordable price.

2. https://www.youtube.com/watch?v=f_6QQ6IVa6E: The woman holding the book stepped onto the patio and looked up to the sky. She was

| Question | Strategy | Question | Strategy |
|---|---|---|---|
| Was there any expert (person or company) (not celebrity) encouraging to use the product/brand? | Authority | Did the video show any normal customers (non-expert, non-celebrity) using the product? | Social Identity |
| Did the video showcase any awards or long usage history of the product/brand? | Authority | Did the video show any customer reviews or testimonials? | Social Proof |
| Was the product/brand comparing itself with other competitors or existing solutions? | Anchoring and Comparison | Were any number/statistics mentioned? | Concreteness |
| Did the video talk about any specific features or provide information about the product/brand? | Concreteness | Were there any mention of any offers on the brand/product? | Reciprocity |
| Were the offers limited or available for a short period of time? | Scarcity | Was the product/brand told to be free or available on a discount? | Foot in the Door, Reciprocity |
| Was the brand/product described as simple, easy-to-use, can start using with minimal resistance? | Overcoming Reactance, Foot in the Door | Was the brand/product talking about bigger societal impact? | Social Impact |
| Did the brand provide any guarantees which might help reduce risk of people to try out the product? | Overcoming Reactance | Did the video provide any resources, tips, guides, or tools related to the product? | Reciprocity |
| When a brand or product is portrayed as human-like? | Anthropomorphism | | |

Table 6: The questions we asked to the non-expert annotators to help them identify persuasion strategy contained in the video advertisement.

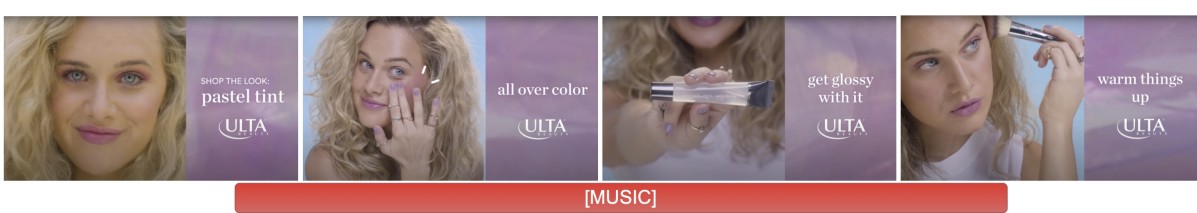

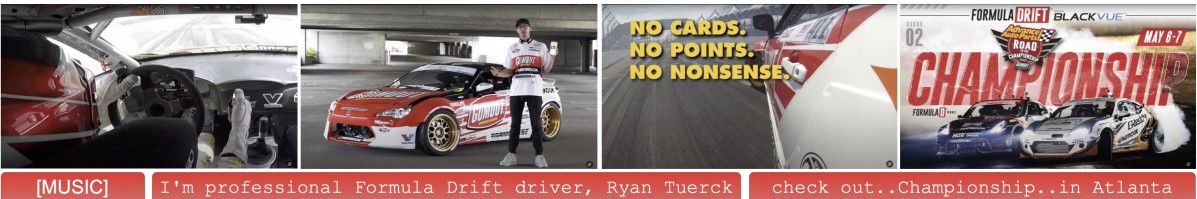

Figure 5: A few examples of stories generated with the proposed pipeline. ASR outputs are shown in Red. As can be seen, the generated stories capture information about scenes, characters, event sequences, dialogues, emotions, and the environment. This helps the downstream models to get adequate information about the video to reason about it correctly. These videos can be watched at https://bit.ly/42Hvq2f, https://bit.ly/3pGcSRr.

ready to take on the day. Taking out her phone, she opened the furniture catalog app, scrolling through the various designs. She quickly decided on the perfect pieces to brighten up her home. Next, she headed to The Home Depot for the supplies she needed. As she entered the store, the woman was delighted to find all the tools and materials she needed, from the Ryobi Cordless Vacuum to the Leaf Blower. She was even more excited when she spotted the Splatter an object recognition tool that allowed her to easily find the perfect paint color for her project. With her shopping done, the woman made her way to the checkout line with a cup of coffee in hand. She couldn't wait to get to work and make her home more beautiful. She knew that with the help of The Home Depot, Today was the Day for Doing. This advertisement was for The Home Depot - the one-stop-shop for all your home improvement needs.

3.     https://www.youtube.com/watch?v=PJlHiQJBDMw: The advertisement for the company Sherwin-Williams opens on a kitchen table strewn with shells and wicker baskets, with two glasses of iced water beside them. A vase with a blue pattern sits in the foreground, and a person holds

up a phone with the Sherwin-Williams logo on the screen. A girl appears from behind a white sheet, peeking out of a white tent as if to signify the timelessness and neutrality of this color. The voice-over begins, as the camera pans to a living room with a staircase, and then to a dining room with a white table, chairs, and a white vase. The words "Color of the Month: Shell White, Sherwin-Williams" appear on the screen, as the camera zooms in on the vase. The words are followed by Öur app makes it a snap,r̈eferring to Color Snap, the company's new way of painting a home. The advertisement ends with the Sherwin-Williams logo, emphasizing the company's commitment to excellence in home painting. This advertisement was promoting the company's color of the month, Shell White.

4.     https://www.youtube.com/watch?v=CDjBIt70fp4: The story began with a green light glowing in the dark, symbolizing the presence of a powerful technology that can change the way we work. This technology was an advanced graphics card, the NVIDIA Quadro FX 1700. It was compared side-by-side with its successor, the Quadro 2000, and it was clear that the Quadro

2000 was far more powerful. The Quadro FX 1700 had a qt - x700 vs qt - x700 capacity, while the Quadro 2000 had a green light that shone brighter and further than before. As the comparison was being made, the results were clear: the Quadro 2000 was the superior product. This advertisement for the company NVIDIA showcased the power of the Quadro 2000, and the improved performance it could bring to an organization. The advertisement concluded with a green light, signaling that NVIDIA had the answer to improving workflows. The product the advertisement was about was the NVIDIA Quadro 2000."

5. [https://www.youtube.com/watch?v=SXTSJC5NrLs](https://www.youtube.com/watch?v=SXTSJC5NrLs): Mary Kom was training hard to make her mark as a boxer, and she had nothing but her focus and determination to help her succeed. She was a force to be reckoned with in the boxing ring and her hard work was soon paying off.But she knew that she had to take her training to the next level if she wanted to win the boxing championship. That's when she decided to take her own advice and step out of her comfort zone. She headed out to the woods, determined to make the most of her time in nature.The woods were a perfect place for Mary to challenge herself physically. She was soon standing on the rocks of a river, pushing her body to its limits with every movement. She was standing tall, her black top glinting in the sunlight, and her energy was palpable. The sun was setting, and it was the perfect time to relax and unwind.Finally, she ended her day with a yoga routine, her body twisting and turning in a beautiful display of strength and grace. It was a day of hard work and fun, and it left Mary feeling inspired and ready to take on anything. The video was an advertisement for the movie Mary Kom, released by Viacom18 Studios."

### A.3 Hallucinations Present In the Automatically Generated Stories

The current large language models , do hallucinate or produce imaginary outputs. However, they are able to capture the gist of the videos, helping improve on downstream tasks. Some examples have been shown in Table-7.

### A.4 Ablation

Among the different components of information input present in the prompt, the LLM utilizes them differently while constructing the verbalization for the videos.For this experiement we use a subset of (Hussain et al., 2017) dataset, considering videos that have spoken audio present.

We use ROUGE-l to get the longest common subsequence (LCS) between the generated verbalization and the individual components,which captures the overlapping content, providing an indication of their semantic similarity.

As generated verbalizations are abstractive as compared to extractive, we also use cosine similarity between the Roberta embeddings of the generated verbalization and the individual components.

We find that despite the order of the components in the prompt, the LLMs tend to utilize the audio components in the videos, in an extractive way.

Table 7: Hallucinations Present In the Automatically Generated Stories

| URL | Story | Hallucinations |
|---|---|---|
| https://youtu.be/_Usyxm9S_Jg | The woman in the advertisement is standing in front of a red background, smiling brightly. She and a man sitting at a table look content, as if they've just come from a conversation. The Netflix logo is visible, indicating that the advertisement is for the company and its products. The voice-over for the advertisement introduces Selena Gomez as the star of the trailer for the popular Twilight Saga. Her popularity has made her a perfect choice to promote Netflix's products. The voice-over then informs viewers that the company's streaming service is not available. The man and woman in the advertisement seem to have a strong connection, and we can infer that they are friends. Perhaps they share an understanding of each other's struggles, since they both seem to be comfortable and happy with each other. The advertisement ends with the Netflix logo, leaving viewers with the knowledge that they can find solace and understanding in the company's products, no matter what they are going through. The advertisement was for Netflix." | The story falsely attributes "Selena Gomez" to "Twilight Saga". The LLMs often confuse between named entity linkages. |
| https://youtu.be/c6ihySYvKYM | The man in glasses was pointing at the Skytech Gaming computer, as he and the man in the headset prepared to start a game. The man with the headset was lost in the world of gaming, his fingers flying across the keyboard as he tried to achieve domination. Meanwhile, the man with glasses watched intently as the RGB TV lit up the room. The woman was playing a game of PC gaming, while the man with glasses and headphones was engrossed in a game of Esports. A man in a headset played a game of PS4, while another man held up a computer screen. A girl was playing a game of PC gaming, while a man was looking at a computer case with colorful lights. The man wearing glasses and headphones was sitting in front of a computer, playing a game of PC gaming. The woman wearing headphones was playing a game on a computer, and the man with glasses was smiling as he played a game of PS4. A person's hand was holding a gaming mouse, while a woman was playing a game of Esports.Finally, a computer case with green lights on it and an Acer Aspire R7 MK2 PC case were in the mix, as the video ended with a man playing a guitar with a lamp and a purple light in the dark. The advertisement was for Skytech Gaming, a company offering gaming PCs and laptops that are more than just a pastime - they are a passion and a battle of the heart. One sentence summarizing the video would be: Skytech Gaming shows that their gaming PCs and laptops can bring out the best of gamers, allowing them to take on every battle with anticipation, adrenaline, and camaraderie. | The entity "Acer Aspire R7 MK2" and "PS4" are not referenced in the prompt and are hallucinated by the LLM |
| https://youtu.be/SZwWT5lnHf8 | A young girl is walking through her bedroom when she notices a man running up the stairs in his sneakers. She is intrigued and follows him, curious to see where he is going. Arriving at the top of the stairs, she finds a young man standing with his backpack. Across the room, she sees a girl in a dress standing in front of a mirror. The girl smiles and looks content. At that moment, the voice-over of the Academy Sports + Outdoors ad can be heard: "Academy Sports and Outdoors is the place to get all the best brands in thousands of combinations for back to school. Right stuff, low price, every day." Suddenly, a black cat can be seen flying in the sky and the girl in the tank top is now standing in front of the mirror. The man is wearing blue pants and sneakers, and the shoes, rug, and sneakers of the apartment are visible. The girl smiles and looks content. The scene shifts to the bedroom where the bed, dresser, and mirror are present, along with a sign for Academy Sportswear. This advertisement for Academy Sports + Outdoors was about the company's back-to-school collection of products. It showcased how Academy Sports + Outdoors provides the right stuff, at a low price, every day, for students to get ready for school. | The scene with a black cat is hallucinated though not mentioned in the prompt |
| https://youtu.be/B63ubKGNqos | The advertisement by Netflix, titled "Many Worlds, One Home", is a stunning vision of many different universes. We see a man standing on a red moon, a desert landscape, and a live performance by Adam Sandler. Purple backgrounds with the word "worlds" convey the idea of many possibilities, while a boy playing with a ball in front of a statue shows a more peaceful and homely side to this. As the night falls, a man is standing on a car, and the Ultraman. There are also glimpses of Tse Seeds, Devilman Crybaby, Saint Seiya: Knights of the Zodiac, Eden Vs. Eden, Carole and Tuesday, Aggretsuko, Seis Manos, Godzilla, Castlevania, B: The Beginning, Baki, Cannon Busters, Rilakkuma and Kaoru, Beastars, and Ultramarine Magmell. We see a teddy bear, knights of the zodiac, a character walking in the water, a man in a snowy area, a pair of shoes, a man running away from the fire, the moon, a man wearing a suit with a green light, eyes being the only thing that can be seen, birds flying in the sky, a red and black background with a red light, bats and castles, the eyes, a man holding a gun, a man flying in the air, a face, a girl with long hair, and two hands. All of this culminates to signify the many different worlds that one can find "home", and the advertisement concludes with the Netflix logo. This advertisement for Netflix captures the idea of many possibilities and of being at home in a single place. | Blip2 hallucinates entities as "Adam Sandler" who is not associated with the advertisement. But it causes the generated story to have the same association. |
| https://youtu.be/WCIEmrN8ot4 | The sun shone brightly in the savanna as the cheetah cub frolicked in the grass. Suddenly, a cheetah emerged from the shadows. The cheetah and it's cub plan to chase the gazelles. Despite the presence of death, life in the wild continued on, just as it had for centuries. This story is featured in the advertisement for Walt Disney Studios, which captures the beauty and unpredictable nature of life in the African savanna. This advertisement ultimately serves to promote the film African Cats, which tells the story of a cheetah family as they strive to protect their respective prides against a variety of obstacles in the wild. | The generated caption "a fox cub is playing in the grass cheetah eats in the wild" was insufficient to capture the overall information of the video. However, due to presence of ASR and world-knowledge of the LLM.The LLM is able to generate a coherent story. |
| https://youtu.be/Lh5n4HcLvCU | The woman in the Ulta Beauty advertisement skillfully applied makeup to her face, creating a beautiful pastel tint. With each brush stroke, her look became brighter and more dreamy. She beamed with confidence and joy, her face glowing with beauty. As the advertisement continued, she highlighted her eyes and cheeks with subtle pastel shades, adding to the stunning effect. With each addition, her face lit up with a vibrant luminosity. When she was finished, her face was a work of art, the perfect combination of bright and dreamy. The Ulta Beauty advertisement showcased the power of a perfect pastel tint, and how it can enhance any look. This advertisement was for Ulta Beauty. | The video contains only music. However, the generated verbalization is able to pick the gist of the advertisement. |
| https://youtu.be/VdhRmVnbqQo | Jackie had been wanting diamond hoop earrings for what seemed like forever. She had been dreaming of the perfect pair that she could wear on special occasions and that would bring out the best in her. One day Jackie happened to come across an advertisement for Costco, which featured a pair of Round Brilliant Diamond Hoop Earrings in 14kt White Gold. She immediately knew this was the perfect pair for her. The advertisement had a voice-over that said, "These earrings sparkle with glamour and sophistication." Jackie was mesmerized by the sparkle of the diamonds, and she was sure anyone who saw them would also be taken in by their beauty. Jackie decided to purchase the earrings. She was sure they would make the perfect accessory for any special occasion. From date nights to family gatherings, she knew these earrings would make her look simply stunning. The advertisement for the Costco Round Brilliant Diamond Hoop Earrings in 14kt White Gold proved to be the perfect choice for Jackie. They not only added a touch of glamour and sophistication to her wardrobe, but made her feel extra special when she wore them. The advertisement for the company Costco was for a pair of Round Brilliant Diamond Hoop Earrings in 14kt White Gold. | The video has no audio transcription, and minimal visual information. However the LLM hallucinates a story, maintaining the essence of the advertisement. It also attributes a false voice-over which is not present in the video. |

| Model | Topic | Emotion | | Persuasion | Action | Reason |
|---|---|---|---|---|---|---|
| | | All labels | Clubbed | | | |
| BLIP-2 Captions + Flant-t5-xxl | 32.2 | 7.4 | 43.11 | 32.1 | 52.98 | 76.26 |
| BLIP-2 Captions + GPT-3.5 | 32.7 | 7.9 | 76.69 | 30.1 | 49.91 | 58.71 |
| Audio Transcription + Flant-t5-xxl | 49.37 | 10.1 | 63.56 | 21.9 | 66.17 | 79.68 |
| Audio Transcription + GPT-3.5 | 32.88 | 6.4 | 75.97 | 32.25 | 64.98 | 61.78 |

Table 8: Ablation study of using only visual (caption) or audio (transcripts) and LLMs for downstream tasks. It can be noted that the overall model does not perform as well (compared to Table 2) when using only audio or scene description without generating story.

Table 9: Comparison of factors contributing to the verbalization

| Model | Component | ROUGE-l | Cosine Similarity |
|---|---|---|---|
| GPT-3.5 | Visual Captions | 18.9 | 50.40 |
| | ASR | 26.29 | 52.60 |
| | OCR | 02.64 | 22.52 |
| Vicuna | Visual Captions | 15.44 | 48.23 |
| | ASR | 21.15 | 48.75 |
| | OCR | 02.63 | 22.52 |

| Model | Top-5 Accuracy | mAP |
|---|---|---|
| VideoMAE | 25.57 | 24.79 |
| InternVideo | 7.477 | 15.62 |
| GPT-3.5 Generated Story + GPT-3.5 | 34.2 | 27.53 |
| Vicuna Generated Story + GPT-3.5 | 31.54 | 27.24 |
| GPT-3.5 Generated Story + Flant5 | 37 | 27.96 |
| Vicuna Generated Story + Flant5 | 31.13 | 27.32 |

Table 10: Top-5 accuracy, and mAP for persuasion strategy detection task

| Method | Frame Extraction | METEOR | CIDEr | Rouge-l | BLEU-1 | BLEU-2 | BLEU-3 | BLEU-4 |
|---|---|---|---|---|---|---|---|---|
| GPT-3.5 | Uniform Sampling | 24.8 | 102.4 | 24.3 | 63.8 | 56.4 | 47.2 | 38.6 |
| GPT-3.5 | Pyscenedetect | 24.17 | 67.8 | 21.17 | 54.59 | 49.05 | 41.54 | 33.88 |

Table 11: Comparison of Pyscenedetect (Breakthrough, 2023) with uniform sampling of choosing video frames. Based on downstream performance, we can see that uniform sampling works better than Pyscenedetect