# OpenReview forum: "A Video Is Worth 4096 Tokens: Verbalize Story Videos To Understand Them In Zero Shot"
_EMNLP/2023/Conference — EMNLP 2023 Main_

### Official Review · Reviewer_Rk8R · 2023-08-01

**Typos Grammar Style And Presentation Improvements:** 1. The caption of Figure 2 lacks a fu…
**Soundness:** 3

**Excitement:**

3: Ambivalent: It has merits (e.g., it reports state-of-the-art results, the idea is nice), but there are key weaknesses (e.g., it describes incremental work), and it can significantly benefit from another round of revision. However, I won't object to accepting it if my co-reviewers champion it.

**Missing References:**

N/A

**Paper Topic And Main Contributions:**

This paper proposes a comprehensive framework that aims to extract and convert information from a lengthy video and generates a corresponding story text in a zero-shot setting. Additionally, the effectiveness of the framework is discussed with respect to the quality of the generated story text and its performance in downstream video understanding tasks, providing evidence of its superiority over previous methods. Furthermore, the paper also presents a classification dataset of advertising video persuasion strategies.

**Questions For The Authors:**

1. "Uniformly Sampled BLIP-2 Captions" in Table 1 has no method description.
2. Why do the three methods (i.e., GPT-3.5, Vicuna, and Flant-t5-XXL) in Table 1 have large performance gaps in multiple metrics?  Is it proportional to the base model capability?
3. Should the downstream tasks in Table 2 add the fine-tuning setting of (Generated story + normal text classifier)? Moreover, why is vicuna's performance so different from the other methods?

**Reasons To Accept:**

1. This paper provides a feasible framework for long video understanding and generation, and describes the design in the framework in detail. The experiment further proves its effect and feasibility.
2. This paper releases a dataset for advertising video persuasion strategy classification.

**Reasons To Reject:**

1. The framework proposed in this paper bears a relatively large similarity to a previous visual understanding framework (e.g. [1]), as it is a common practice to convert visual and audio information into text and then utilize prompts to construct it with input into LLM. Therefore, this paper lacks a sufficiently novel contribution.
2. Video understanding is a commonly undertaken task, and numerous works and datasets have been proposed in recent years. Among these, longer videos are typically seen as more intricate tasks for video understanding, given the characteristic multi-camera transitions and the greater length of the video. Whilst this paper highlights its ability to handle long videos, it does not delve into these aspects above.

[1] Visual Clues: Bridging Vision and Language Foundations for Image Paragraph Captioning, Yujia Xie, Luowei Zhou, Xiyang Dai, Lu Yuan, Nguyen Bach, Ce Liu, Michael Zeng

**Reproducibility:**

3: Could reproduce the results with some difficulty. The settings of parameters are underspecified or subjectively determined; the training/evaluation data are not widely available.

**Reviewer Confidence:**

3: Pretty sure, but there's a chance I missed something. Although I have a good feel for this area in general, I did not carefully check the paper's details, e.g., the math, experimental design, or novelty.

---

> ### Author Rebuttal · Authors · 2023-08-29
>
> **Comparison wrt the Visual Clues, Xie et al paper**
> Thank you for pointing us to the visual clues paper. There are certainly similarities between our work and visual clues. We find that both works differ in the following ways:
> - Focus for both papers is quite different. Visual clues focuses on generating good image paragraph captioning. That’s why they compare their performance with other visual captioning models. We focus on solving downstream tasks for which the video domain doesn’t have many annotated samples to learn a task. We show that we get much better generalization across tasks by converting visual information to stories and then finetuning a language model on the generated text as opposed to finetuning a task-specific vision model on raw videos. For us, good story generation is an intermediary step, for visual clues, that’s the end goal.
>
> - The methods are quite different. While both our paper and visual-clues use captioner and object detectors, visual-clues rely on a visual-language model (like BLIP) to detect which paragraph is the best candidate. We don’t do that. Further, we train an LM (Roberta) on the generated stories to perform downstream tasks better, visual-clues don’t do that. We also show that just concatenating “visual clues” doesn’t perform well on the downstream tasks (Table-4).
>
> - The data that both papers deal with is quite different. Visual clues focuses on single-frame images; we focus on short (<120 seconds) and long (12.4 minutes avg) videos. The videos we deal with have audio, music, speech, on-screen text, fast-moving frames, animations, etc, needing world knowledge. The Visual genome data and the Stanford dataset (Krause et al 2017) contain camera-shot images in the natural surroundings. They contain pictures of monuments, outdoors, indoors, etc. Our videos, on the other hand, are both camera-shot videos (Li et al) and computer-generated graphics (Hussain et al and persuasion strategy datasets).
>
>
>
> **Performance on other datasets:**
> - Reviewer 2 pointed us to the HVU and LVU datasets. Thanks to that, in the limited rebuttal time, we were able to test out our best models on the following tasks across the two datasets. Similar to the tasks reported in the paper, we obtained significant improvements in them.
>
> We observe that on 8/9 tasks in the LVU benchmark and 6/6 tasks in the HVU benchmark, we get state-of-the-art results.
>
>
> 1. LVU Benchmark (https://openaccess.thecvf.com/content/CVPR2021/papers/Wu_Towards_Long-Form_Video_Understanding_CVPR_2021_paper.pdf):
>
> | Training         | Model                                            | relationship | way_speaking | scene | like_ratio | view_count | director | genre | writer | year  |
> |------------------|--------------------------------------------------|--------------|--------------|-------|------------|------------|----------|-------|--------|-------|
> | Trained          | R101-slowfast+NL [ICCV,2019]                     | 52.4         | 35.8         | 54.7  | 0.386      | 3.77       | 44.9     | 53    | 36.3   | 52.5  |
> | Trained          | VideoBert, Sun et al, 2019, ICCV                 | 52.8         | 37.9         | 54.9  | 0.32       | 4.46       | 47.3     | 51.9  | 38.5   | 36.1  |
> | Trained          | Xiao et al, 2022, CVPR                           | 50.95        | 34.07        | 44.19 | 0.353      | 4.886      | 40.19    | 48.11 | 31.43  | 29.65 |
> | Trained          | CVRL, Qian et al, 2021, CVPR                     | 50.95        | 32.86        | 32.56 | 0.444      | 4.6        | 37.76    | 48.17 | 27.26  | 25.31 |
> | Trained          | Object Transformer, Wu et al, 2019, CVPT         | 53.1         | **39.4**         | 56.9  | 0.23       | **3.55**       | 51.2     | 54.6  | 34.5   | 39.1  |
> | Zero-shot (Ours) | GPT-3.5 generated story + Flant5-xxl [zero shot] | 64.1         | **39.07**        | 60.2  | 0.061      | 12.84      | 69.9     | **58.1**  | **52.4**   | 75.6  |
> | Zero-shot (Ours) | GPT-3.5 Generated story + GPT-3.5 classifier     | **68.42**        | 32.95        | 54.54 | **0.031**      | 12.69      | **75.26**    | 50.84 | 32.16  | **75.96** |
> | Trained (Ours)   | GPT-3.5 generated story + Roberta                | 62.16        | 38.41        | **68.65** | 0.054      | 11.84      | 45.34    | 39.27 | 35.93  | 7.826 |
>
> (like ratio and view count are MSE scores, (less is better), all others are top-1 accuracies, (more is better))
>
>
> 2. HVU benchmark (https://pages.iai.uni-bonn.de/gall_juergen/download/HVU_eccv20.pdf)
> | Training  | Model                             | Scene | Object | Action | Event | Attribute | Concept | Overall |
> |-----------|-----------------------------------|-------|--------|--------|-------|-----------|---------|---------|
> | Trained   | 3D-Resnet, Diba et al, 2020, ECCV | 50.6  | 28.6   | 48.2   | 35.9  | 29        | 22.5    | 35.8    |
> | Trained   | 3D-STCNet, Diba et al, 2020, ECCV | 51.9  | 30.1   | 50.3   | 35.8  | 29.9      | 22.7    | 36.7    |
> | Trained   | HATNet, Diba et al, 2020, ECCV    | 55.8  | 34.2   | 51.8   | 38.5  | 33.6      | 26.1    | 40      |
> | Trained | 3D-Resnet, Diba et al, 2020, ECCV [Multitask] | 51.7 | 29.6 | 48.9 | 36.6 | 31.1 | 24.1 | 37   |
> | Trained | HATNet, Diba et al, 2020, ECCV [Multitask]    | 57.2 | 35.1 | 53.5 | **39.8** | 34.9 | 27.3 | 41.3 |
> | Trained   | GPT-3.5+Flant5 | **59.66** | **98.89** | **98.96** | 38.42 | **67.76** | **86.99** | **75.12** |
> | Zero-Shot | GPT-3.5+GPT3.5 | **60.2**  | **99.16** | **98.72** | **40.79** | **67.17** | **88.6**  | **75.77** |
>
> (All the tasks are MAP scores, (more is better))
>
> **Method description of "Uniformly Sampled BLIP-2 Captions"**
> - We sample every 10th frame at native fps of the video and generate captions through Blip2. These captions are then deduplicated among themselves, and consecutive captions having 80% word overlap are dropped.
> - This final set of captions is used to prompt the LLM to generate verbalisation for the video.
>
> **The three methods of GPT-3.5, Vicuna, and Flant-t5-XXL having large performance gaps in multiple metrics. More discussion of Vicuna’s results:**
> - GPT-3.5, because of its bigger size (13x bigger), generates better stories than Vicuna. We observe this across all models.
> - We find that Flan-t5 generally performs better than other models for all classification tasks. Though this phenomenon is new and we would like to investigate this phenomenon more, but this is consistent with a few other papers that have observed similar results:
> https://arxiv.org/abs/2005.14165 - they observe that in several tasks, other models beat GPT-3 (various sized) models in few-shot settings. - Further, in some tasks, smaller models outperform larger models
> - https://arxiv.org/abs/2204.03954 - they observe that in-context learning based GPT-3 does not achieve higher results than much smaller models like BERT.
> - Vicuna, as a classifier, performs worse than all other methods.
>
> **The fine-tuning setting of (Generated story + normal text classifier) in Table 2**
> - We have shown the performance of Roberta-base trained for the classification tasks in Table-2 ( Generated Story + Roberta Classifier ).

---

### Official Review · Reviewer_5637 · 2023-08-04

**Soundness:** 4

**Excitement:**

4: Strong: This paper deepens the understanding of some phenomenon or lowers the barriers to an existing research direction.

**Missing References:**

Some relevant references that are missing:

[1] **LVU dataset**: https://openaccess.thecvf.com/content/CVPR2021/papers/Wu_Towards_Long-Form_Video_Understanding_CVPR_2021_paper.pdf

[2] **HVU dataset**: https://pages.iai.uni-bonn.de/gall_juergen/download/HVU_eccv20.pdf

[3] **LLAVa**: https://arxiv.org/pdf/2304.08485.pdf

**Paper Topic And Main Contributions:**

The paper presents a zero-shot method for video understanding by extracting long coherent descriptions through various large language models.  The major contributions of this paper are:
* Usage of various multimodal models to extract textual information
* Aggregation of textual information through large language models followed by text-based reasoning/classification tasks.

**Questions For The Authors:**

* For **BLIP2** caption extraction, keyframe selection was used as a pre-filtering step.
As an alternative, have the authors explored shot detection followed by BLIP-2-based caption extraction for center frames in shots?

* For the persuasion strategy dataset zero-shot results, it is not clearly mentioned that the evaluation was performed on the complete set of 1002 videos or a separate test split.

**Reasons To Accept:**

* Multi-stream extraction of textual information through various multimodal models, including scene descriptions, embedded text (OCR), and transcripts (speech).
* Aggregation of multiple forms of text information through large-language models for generating a coherent description.
* Usage of coherent descriptions for multiple media-centered video-understanding tasks

**Reasons To Reject:**

* Hallucination examples are provided, but failure examples of individual pipeline components and how they were handled are not mentioned/shown
     * For example, where **BLIP-2** fails to provide sufficient captions for the keyframes
     * Lack of speech and only music resulting in no transcription
* Details regarding the persuasion strategy dataset are missing wrt following:
     * How were the brands selected?
     * Details regarding the pool of human annotators, including how many unique annotators were used for the entire task?
     * In the case of the multi-label nature of persuasion strategies, top-5 accuracy, and **mAP** should be reported as metrics for comparing different methods (**Table 2**)
* Experimental details:
     * Finetuning details regarding the usage of **VideoMAE** and **InternVideo** are lacking in Section 4.2
* To test the utility of the proposed method as a purely zero-shot alternative for video description-driven understanding, comparisons should be shown on complex media-based video tasks, including **LVU** and **HVU**.

**Reproducibility:**

4: Could mostly reproduce the results, but there may be some variation because of sample variance or minor variations in their interpretation of the protocol or method.

**Reviewer Confidence:**

4: Quite sure. I tried to check the important points carefully. It's unlikely, though conceivable, that I missed something that should affect my ratings.

**Typos Grammar Style And Presentation Improvements:**

* Overall, the paper is well written with clear motivation.
* If dataset names are mentioned in the table captions, It will be easier to understand the improvements wrt different tasks.
     *  In **Table 2**, it should be clearly mentioned that the persuasion strategy section is from the in-house collected dataset, whereas the emotion, topic and action+ tasks are associated with the advertisement dataset from Hussain et.al

---

> ### Author Rebuttal · Authors · 2023-08-29
>
> **Performance on other datasets:**
> - Thanks for pointing us to the LVU and HVU datasets! In the limited rebuttal time, we were able to test out our best models on the following tasks across the two datasets. Similar to the tasks reported in the paper, we obtain significant improvements on them.
> We observe that on 8/9 tasks in the LVU benchmark and 6/6 tasks in the HVU benchmark, we get state-of-the-art results.
> We observe that on 8/9 tasks in the LVU benchmark and 6/6 tasks in the HVU benchmark, we get state-of-the-art results.
>
>
> 1. LVU Benchmark (https://openaccess.thecvf.com/content/CVPR2021/papers/Wu_Towards_Long-Form_Video_Understanding_CVPR_2021_paper.pdf):
>
> | Training         | Model                                            | relationship | way_speaking | scene | like_ratio | view_count | director | genre | writer | year  |
> |------------------|--------------------------------------------------|--------------|--------------|-------|------------|------------|----------|-------|--------|-------|
> | Trained          | R101-slowfast+NL [ICCV,2019]                     | 52.4         | 35.8         | 54.7  | 0.386      | 3.77       | 44.9     | 53    | 36.3   | 52.5  |
> | Trained          | VideoBert, Sun et al, 2019, ICCV                 | 52.8         | 37.9         | 54.9  | 0.32       | 4.46       | 47.3     | 51.9  | 38.5   | 36.1  |
> | Trained          | Xiao et al, 2022, CVPR                           | 50.95        | 34.07        | 44.19 | 0.353      | 4.886      | 40.19    | 48.11 | 31.43  | 29.65 |
> | Trained          | CVRL, Qian et al, 2021, CVPR                     | 50.95        | 32.86        | 32.56 | 0.444      | 4.6        | 37.76    | 48.17 | 27.26  | 25.31 |
> | Trained          | Object Transformer, Wu et al, 2019, CVPT         | 53.1         | **39.4**         | 56.9  | 0.23       | **3.55**       | 51.2     | 54.6  | 34.5   | 39.1  |
> | Zero-shot (Ours) | GPT-3.5 generated story + Flant5-xxl [zero shot] | 64.1         | **39.07**        | 60.2  | 0.061      | 12.84      | 69.9     | **58.1**  | **52.4**   | 75.6  |
> | Zero-shot (Ours) | GPT-3.5 Generated story + GPT-3.5 classifier     | **68.42**        | 32.95        | 54.54 | **0.031**      | 12.69      | **75.26**    | 50.84 | 32.16  | **75.96** |
> | Trained (Ours)   | GPT-3.5 generated story + Roberta                | 62.16        | 38.41        | **68.65** | 0.054      | 11.84      | 45.34    | 39.27 | 35.93  | 7.826 |
>
> (like ratio and view count are MSE scores, (less is better), all others are top-1 accuracies, (more is better))
>
>
> 2. HVU benchmark (https://pages.iai.uni-bonn.de/gall_juergen/download/HVU_eccv20.pdf)
> | Training  | Model                             | Scene | Object | Action | Event | Attribute | Concept | Overall |
> |-----------|-----------------------------------|-------|--------|--------|-------|-----------|---------|---------|
> | Trained   | 3D-Resnet, Diba et al, 2020, ECCV | 50.6  | 28.6   | 48.2   | 35.9  | 29        | 22.5    | 35.8    |
> | Trained   | 3D-STCNet, Diba et al, 2020, ECCV | 51.9  | 30.1   | 50.3   | 35.8  | 29.9      | 22.7    | 36.7    |
> | Trained   | HATNet, Diba et al, 2020, ECCV    | 55.8  | 34.2   | 51.8   | 38.5  | 33.6      | 26.1    | 40      |
> | Trained | 3D-Resnet, Diba et al, 2020, ECCV [Multitask] | 51.7 | 29.6 | 48.9 | 36.6 | 31.1 | 24.1 | 37   |
> | Trained | HATNet, Diba et al, 2020, ECCV [Multitask]    | 57.2 | 35.1 | 53.5 | **39.8** | 34.9 | 27.3 | 41.3 |
> | Zero-Shot (Ours)  | GPT-3.5+Flant5 | **59.66** | **98.89** | **98.96** | 38.42 | **67.76** | **86.99** | **75.12** |
> | Zero-Shot (Ours) | GPT-3.5+GPT3.5 | **60.2**  | **99.16** | **98.72** | **40.79** | **67.17** | **88.6**  | **75.77** |
>
> (All the tasks are MAP scores, (more is better))
>
>
> **Details regarding the persuasion strategy dataset:**
> - While we tried to accommodate the most important details in the 8 pages, we apologize for missing out on some key details. We will make sure to add these details in the final version. The details are:
>     - Video and Brands selection - 155 Fortune 500 brands, covering 113 industries, the ads spanned the years 2008-2022. The videos have an avg duration of 33 secs. Nearly 45% of the videos have audio in them. The collected advertisement videos have a variety of characteristics, including different scene velocities, human presence and animations, visual and audio branding, a variety of emotions, visual and scene complexity, and audio types.
>     - Unique annotators - 484
>     - Annotators were university students who were given guidelines to annotate videos and were duly compensated for their work based on local rates.
>     - Top-5 accuracy and MAP -
> | Model                            | Top-5 Accuracy | mAP   |
> |----------------------------------|----------------|-------|
> | VideoMAE                         | 25.57          | 24.79 |
> | InternVideo                      | 7.477          | 15.62 |
> | GPT-3.5 Generated Story+ GPT-3.5 | 34.2           | 27.53 |
> | Vicuna Generated Story+GPT-3.5   | 31.54          | 27.24 |
> | GPT-3.5 Generated Story+Flant5   | 37             | 27.96 |
> | Vicuna Generated Story+Flant5    | 31.13          | 27.32 |
>
>
>
> **Experimental details:**
> - Finetuning details of VideoMAE and InternVideo -
>     - VideoMAE : We finetune VideoMAE with an MLP head of n-way classification task (n depending on the number of labels). We divide each video into 13 non-overlapping parts, and sample 16 frames at 2fps (taken from the VideoMAE paper). Each part of the video is input to VideoMAE to get an embedding of (1,768). Each individual embedding is concatenated to form a (13,768) embedding, which becomes the representation of the video and is input to the MLP head. The MLP has a MaxPoolLayer and 2 fully connected Linear layers. We use ReLU activation to train the model.VideoMAE weights are frozen during training.
>
>     - InternVideo :  We finetune InternVideo with an MLP head of n-way classification task (n depending on the number of labels). The InternVideo returns a (1,768) embedding representation of the video. We use an MLP with 2 fully connected linear layers to finetune for the tasks.
>
>
>
>
> **Shot detection method**
> - We agree that our current method of uniform sampling might discard a few important keyframes. Experimentally, we observe that:
>     - There is a lot of information redundancy in the video: the same/complementary information is repeated through audio, video frames, OCR, etc. We find that removing video frames completely did not alter the performance on the downstream benchmarks significantly. The numbers w/o visual information are given in Table 4 (Ablation)
>     - Uniform sampling captures the most important information. Our sampling rate is 2-3 fps (depending on the native video fps). This might miss a few frames but it captures most important ones.
>     - For shorter videos (with length than <120 seconds), we use GMFlow-based sampling algorithm (Fig.2, Sec.3.1 Line 292).
>     - However, we agree with the reviewer’s new experiment. In the limited rebuttal time, we performed an additional experiment where we compare the verbalisations generated on Video storytelling dataset, we find that using pyscenedetect (https://github.com/Breakthrough/PySceneDetect) the scores were comparable. We will perform more experiments and put them in the camera-ready.
> |Method|Frame Extraction|METEOR|CIDEr|Rouge-l|Bleu-1|Bleu-2|Bleu-3|Bleu-4|
> |:----|:----|:----|:----|:----|:----|:----|:----|:----|
> |GPT-3.5|Uniform Sampling|24.8|102.4|24.3|63.8|56.4|47.2|38.6|
> |GPT-3.5|Pyscenedetect|24.17|67.8|21.17|54.59|49.05|41.54|33.88|
>
>
>
>
> **Persuasion strategy dataset zero-shot results, test settings**
> - For zero shot results, we perform evaluation on 1002 videos. For finetuning, we perform evaluation on a test split of 198 videos. The same test-split is used for VideoMAE, InternVideo and Roberta Classifier.
>
>
> **Hallucination and failure examples**
> - For example, where BLIP-2 fails to provide sufficient captions for the keyframes
>     - **Example 1**
>     - Video_url = https://www.youtube.com/watch?v=B63ubKGNqos
>     - Deduplicated BLIP Caption =
>         - a man standing on a red moon many
>         - a desert landscape many
>         - a saturday night live performance worlds
>         - a purple background with the word worlds one by adam sandler one
>         - a character in a black background a boy is playing with a ball in front of a statue a man is standing on a car at night ultraman
>         - ad-screenshot nascar driver nascar nascar nascar n a character's face with a black eye tse seeds
>         - tse seeds devilman cybaby devilman cybaby saint sevyx knights of the zodiac eden vs eden carole and tuesday aggretsu a cartoon character is flying in the sky a woman is holding a satchel se manos - el juego de juegos godzilla
>         - hd wallpaper a man is standing in the snow with a torch Castlevania
>         - a saber-toothed tiger the beginning a girl with big eyes canon busters rilakuma and koru rilakuma and koruru a dragon is standing in front of a blue light ultramarine magnell knights of the zodiac 2 global netflix logo
>
>     - Generated Story =
>         > The advertisement by Netflix, titled \"Many Worlds, One Home\", is a stunning vision of many different universes. We see a man standing on a red moon, a desert landscape, and a live performance by Adam Sandler. Purple backgrounds with the word \"worlds\" convey the idea of many possibilities, while a boy playing with a ball in front of a statue shows a more peaceful and homely side to this. As the night falls, a man is standing on a car, and the Ultraman. There are also glimpses of Tse Seeds, Devilman Crybaby, Saint Seiya: Knights of the Zodiac, Eden Vs. Eden, Carole and Tuesday, Aggretsuko, Seis Manos, Godzilla, Castlevania, B: The Beginning, Baki, Cannon Busters, Rilakkuma and Kaoru, Beastars, and Ultramarine Magmell. We see a teddy bear, knights of the zodiac, a character walking in the water, a man in a snowy area, a pair of shoes, a man running away from the fire, the moon, a man wearing a suit with a green light, eyes being the only thing that can be seen, birds flying in the sky, a red and black background with a red light, bats and castles, the eyes, a man holding a gun, a man flying in the air, a face, a girl with long hair, and two hands. All of this culminates to signify the many different worlds that one can find \"home\", and the advertisement concludes with the Netflix logo. This advertisement for Netflix captures the idea of many possibilities and of being at home in a single place.
>
>     - Remark = Blip2 hallucinates entities as “Adam Sandler” who is not associated with the advertisement. But it causes the generated story to have the same association.
>
>
>     - **Example 2**
>     - Video URL = https://www.youtube.com/watch?v=WCIEmrN8ot4
>     - Deduplicated BLIP Captions = a fox cub is playing in the grass cheetah eats in the wild
>     - Generated Story =
>         > The sun shone brightly in the savanna as the cheetah cub frolicked in the grass. Suddenly, a cheetah emerged from the shadows. The cheetah and it's cub plan to chase the gazzeles. Despite the presence of death, life in the wild continued on, just as it had for centuries. This story is featured in the advertisement for Walt Disney Studios, which captures the beauty and unpredictable nature of life in the African savanna. This advertisement ultimately serves to promote the film African Cats, which tells the story of a cheetah family as they strive to protect their respective prides against a variety of obstacles in the wild.
>     - Remarks = The captions are insufficient to capture the overall information of the video. However, due to presence of ASR and world-knowledge of the LLM.The LLM is able to generate a coherent story.
>
> - For example, Lack of speech and only music resulting in no transcription
>     - **Example 1**
>     - Video URL = https://www.youtube.com/watch?v=Lh5n4HcLvCU
>     - Generated Story =
>        > The woman in the Ulta Beauty advertisement skillfully applied makeup to her face, creating a beautiful pastel tint. With each brush stroke, her look became brighter and more dreamy. She beamed with confidence and joy, her face glowing with beauty. As the advertisement continued, she highlighted her eyes and cheeks with subtle pastel shades, adding to the stunning effect. With each addition, her face lit up with a vibrant luminosity. When she was finished, her face was a work of art, the perfect combination of bright and dreamy. The Ulta Beauty advertisement showcased the power of a perfect pastel tint, and how it can enhance any look. This advertisement was for Ulta Beauty.
>     - Remark = The video contains only music. However, the generated verbalisation is able to pick the gist of the advertisement.
>
>    - **Example 2**
>    - Video URL = https://www.youtube.com/watch?v=VdhRmVnbqQo
>    - Generated story =
>         > Jackie had been wanting diamond hoop earrings for what seemed like forever. She had been dreaming of the perfect pair that she could wear on special occasions and that would bring out the best in her. One day Jackie happened to come across an advertisement for Costco, which featured a pair of Round Brilliant Diamond Hoop Earrings in 14kt White Gold. She immediately knew this was the perfect pair for her. The advertisement had a voice-over that said, “These earrings sparkle with glamour and sophistication." Jackie was mesmerized by the sparkle of the diamonds, and she was sure anyone who saw them would also be taken in by their beauty. \n\nJackie quickly decided to purchase the earrings. She was sure they would make the perfect accessory for any special occasion. From date nights to family gatherings, she knew these earrings would make her look simply stunning. The advertisement for the Costco Round Brilliant Diamond Hoop Earrings in 14kt White Gold proved to be the perfect choice for Jackie. They not only added a touch of glamour and sophistication to her wardrobe, but made her feel extra special when she wore them. \n\nThe advertisement for the company Costco was for a pair of Round Brilliant Diamond Hoop Earrings in 14kt White Gold.
>     - Remark = The video has no audio transcription, and minimal visual information. However the LLM hallucinates a story, maintaining the essence of the advertisement. It also attributes a false voice-over which is not present in the video.

---

### Official Review · Reviewer_Jhzm · 2023-08-10

**Soundness:** 4

**Excitement:**

4: Strong: This paper deepens the understanding of some phenomenon or lowers the barriers to an existing research direction.

**Paper Topic And Main Contributions:**

They are leveraging the powerful understanding and reasoning capabilities of LLMs to generate a story from a usually long video which will be used to do video understanding tasks like: emotions depicted and persuasion strategies used.
There is a modality gap between videos consisting of images, audios, texts and etc... and text that is primary input of an LLM. They address this issue by converting the video to text which will be fed to an LLM to generate a story that is used in video understanding tasks instead of the raw video.
When it comes to contributions they specifically did:
1) Converting long videos from the multimodal domain to “small” coherent textual stories by verbalizing keyframes, audio, and text-overlaid scenes with the help of a powerful LLM and instructions. In their experiments, the story generation capability of their method outperforms state-of-the-art story generation methods
2) Test the utility of generated stories by conducting extensive experiments on several benchmark datasets covering five video understanding tasks, namely topic, emotion and persuasion strategy classification, action and reason retrieval and generation. Experimental results demonstrate that their method achieve better results than both finetuned and zero-shot video understanding baseline models without using any human-annotated samples.
3) For the first time in literature, they show that the essence of a highly-multimodal video can be represented in text while being informed through the different modalities like audio, raw pixels of frames, text overlaid on scenes, emotions, and product and business information.
4) Release the first dataset for studying persuasion strategies in advertisement videos. This enables initial progress on the challenging task of automatically understanding the messaging strategies conveyed through video advertisements.

**Reasons To Accept:**

The paper uses an original methodology in video understanding for the first time which pass other state of the art methods.
The paper is written in clear language.
The author did a through literature study and comparison of results with those studies.

**Reasons To Reject:**

My concerns have been addressed in the rebuttal.

**Reproducibility:**

4: Could mostly reproduce the results, but there may be some variation because of sample variance or minor variations in their interpretation of the protocol or method.

**Reviewer Confidence:**

4: Quite sure. I tried to check the important points carefully. It's unlikely, though conceivable, that I missed something that should affect my ratings.

---

> ### Author Rebuttal · Authors · 2023-08-29
>
> **Keyframe sampling**
>
> We agree that our current method of uniform sampling might discard a few important keyframes. Experimentally, we observe that:
> - There is a lot of information redundancy in the video: similar information is repeated through audio, video frames, OCR, etc. We find that removing video frames completely did not alter the performance on the downstream benchmarks significantly. The numbers w/o visual information are given in Table 4 (Ablation)
> - Uniform sampling captures most important information. Our sampling rate is 2-3 fps (depending on the native video fps). This might miss a few frames but it captures most important ones.
> - For shorter videos (with length than <120 seconds), we use GMFlow based sampling algorithm (Fig.2, Sec.3.1 Line 292).
>
> However, we agree with the reviewer’s new experiment. In the limited rebuttal time, we performed an additional experiment where we compare the verbalisations generated on Video storytelling dataset, we find that using pyscenedetect (https://github.com/Breakthrough/PySceneDetect) the scores were comparable.
>
> |Method|Frame Extraction|METEOR|CIDEr|Rouge-l|Bleu-1|Bleu-2|Bleu-3|Bleu-4|
> |:----|:----|:----|:----|:----|:----|:----|:----|:----|
> |GPT-3.5|Uniform Sampling|24.8|102.4|24.3|63.8|56.4|47.2|38.6|
> |GPT-3.5|Pyscenedetect|24.17|67.8|21.17|54.59|49.05|41.54|33.88|
>
> We will perform more experiments and put them in the camera-ready.
>
> **The reason behind the selection of BLIP-2 model over other VQA models:**
> For our downstream tasks, we needed two main sources of visual information: captions and the objects present in the video.
> We pose both the tasks as VQA (open-ended generation), Blip2 with Flant5 decoder beats all models such as OSCAR, VinVL, Flamingo. Ref: Table-3, Table-4 of BLIP-2:Bootstrapping Language-Image Pre-training with Frozen Image Encoders and Large Language Models (https://arxiv.org/abs/2301.12597).
> Seeing the performance reported, we went with Blip2 as our choice of model, for extraction of visual information.

---

### Meta-Review · Area_Chair_c9tQ · 2023-09-27

**Recommendation:** 5

**Metareview:**

This is an interesting paper that did extensive experimentation on using LLMs for zero-shot video understanding tasks, showing an interesting result beating supervised baselines. They demonstrated the utility of language generation systems in domains where a strong data signal could be limited such as video-based data that heavily relies on human annotation for quality control.

Pros:
- Clear description of their framework, pipeline and task formulation. Would be essential for reproducing such a setup on other tasks
- Their method of multi source text data extraction and comprehension via LLMs is sound
- Experiments are sufficiently detailed and paper is well written with extensive literature review

Cons:
- Representing non-text modalities via text has been covered by other papers, reducing novelty of the idea
- Video understanding is an umbrella term comprising several complex understanding tasks, handling missing data, and varying noise levels/durations of data. Tasks covered in this paper do not necessarily reflect takeaways towards complex video understanding problems.

---

### Decision · Program_Chairs · 2023-10-07

**Decision:**

Accept-Main

**Comment:**

This is an interesting paper that did extensive experimentation on using LLMs for zero-shot video understanding tasks, showing an interesting result beating supervised baselines. They demonstrated the utility of language generation systems in domains where a strong data signal could be limited such as video-based data that heavily relies on human annotation for quality control.

Pros:
- Clear description of their framework, pipeline and task formulation. Would be essential for reproducing such a setup on other tasks
- Their method of multi source text data extraction and comprehension via LLMs is sound
- Experiments are sufficiently detailed and paper is well written with extensive literature review

Cons:
- Representing non-text modalities via text has been covered by other papers, reducing novelty of the idea
- Video understanding is an umbrella term comprising several complex understanding tasks, handling missing data, and varying noise levels/durations of data. Tasks covered in this paper do not necessarily reflect takeaways towards complex video understanding problems.